# Experiences of Caring for Cohort-Isolated Patients among Nurses in Locked Psychiatric Units

**DOI:** 10.3390/healthcare11192650

**Published:** 2023-09-28

**Authors:** Hyeran An, Kyungmi Kim, Jongeun Lee, Sunhwa Won

**Affiliations:** 1Department of Nursing, College of Nursing, Daegu Catholic University, Daegu 42472, Republic of Korea; ahr777@cu.ac.kr; 2Department of Nursing, Gangdong University, Eumseong-gun 27690, Republic of Korea; 96yasi@gangdong.ac.kr; 3Department of Nursing Science, College of Medicine, Chungbuk National University, Cheongju-si 28644, Republic of Korea; 4Graduate School, Chungnam National University, Daejeon 34134, Republic of Korea; 202060288@o.cnu.ac.kr

**Keywords:** psychiatric nursing, mental patients, COVID-19 pandemic, qualitative research

## Abstract

The aim of this study was to gain an understanding of the experiences of caring for cohorts of patients isolated due to coronavirus disease (COVID-19) among nurses in locked psychiatric units. A phenomenological approach was used to analyze data collected from 10 nurses with a minimum of one year of experience as a mental health nurse working in locked psychiatric units that were cohort-isolated due to COVID-19. Data collected using semi-structured one-to-one in-depth interviews were analyzed based on steps outlined by Colaizzi. Five themes and thirteen subthemes emerged. The five themes were as follows: “Challenges intensified by the nature of mental disorders”, “Concerns regarding limited mental health care”, “Escalating stress”, “Bolstered identity as a mental health nurse”, “Witnessing changes that have begun”. Environmental and institutional measures need to be implemented to identify the potential phenomena that may affect locked psychiatric units during an infectious disease epidemic and ensure the safety of health care professionals and patients from the infectious disease.

## 1. Introduction

Mental disorders have been considered a major risk factor for infection even prior to the coronavirus disease (COVID-19) pandemic [1]. Infection control for patients with mental disorders is hindered by not only their psychiatric symptoms or emotional problems but also their diminished comprehension resulting from their distorted perception of reality, communication problems, and lack of capacity for performing health-related activities [2]. The therapeutic environment of locked psychiatric units includes measures such as locked doors and windows due to concerns regarding eloping or self-harm and lack of hand sanitizers in these units due to concerns regarding accidental patient ingestion [3]. Furthermore, psychiatric wards are designed to facilitate close social interaction among patients and staff, which creates a potential for rapid transmission and subsequent outbreaks should one patient or staff member contract an emerging infectious disease [4]. In February 2020, 102 out of 104 patients in a closed psychiatric unit in Gyeongbuk, South Korea, were diagnosed with COVID-19 [3]. Similar cases of COVID-19 outbreaks among patients with mental disorders and health care workers have been reported in mental health institutions globally, including Wuhan, China [5], and Washington, United States [6]. These incidents underscore the vulnerability of individuals with mental illnesses and mental health facilities to disasters, such as infectious disease outbreaks.

Due to mass COVID-19 outbreaks within mental health facilities, mental health professionals have been reported to face an exhaustive workload, resulting in burnout and elevated stress levels [7]. Notably, nurses on the frontline of COVID-19 care [8] are particularly susceptible to adverse psychological outcomes. Failure to adequately diagnose and address these repercussions can lead to long-lasting challenges [9]. However, amidst the challenges, nurses have been observed to experience personal growth and positive transformations as they save lives and address the health issues of patients [10]. Hence, there is a need to explore the essence of this phenomenon of patient care experiences among nurses in cohort-isolated locked psychiatric units through a phenomenological approach. This approach allows researchers to explore the phenomenon of interest without bias [11], so the findings of this study should be utilized to establish the basis of a support system for nurses who care for patients with mental disorders who are highly vulnerable to emerging infectious diseases. Nonetheless, studies on the experiences of nurses caring for COVID-19 patients in Korea and abroad have primarily explored the experiences of nurses in COVID-19 units [12,13,14,15,16,17]. There has been research on the experiences of nurses in parent–child isolation units in COVID-19-designated hospitals [18], but it is difficult to find research on the experiences of nurses in locked psychiatric units with cohort isolation. Given that mental health nursing requires higher expertise and commitment compared to other areas of nursing [19], inadequately trained and educated personnel cannot provide adequate and timely care to patients with mental disorders [7]. Exploring the essence of experiences of patient care among nurses in cohort-isolated locked psychiatric units and establishing a support system for these nurses by ameliorating the environment and systems in locked psychiatric units will be the first steps to securing stable mental health nursing staffing and ensuring their safety and health.

This study investigates the patient care experiences of nurses in locked psychiatric units that were cohort-isolated due to COVID-19 in order to present foundational data to identify the potential phenomena that may occur in locked psychiatric units during an infectious disease outbreak and help nurses to protect themselves and their patients from infection.

## 2. Methods

### 2.1. Design

This phenomenological study aimed to explore the patient care experiences of nurses in locked psychiatric units that were cohort-isolated due to COVID-19. This study was conducted in accordance with the consolidated criteria for reporting qualitative research (COREQ) guidelines [20].

### 2.2. Participant Recruitment

This study included 10 mental health nurses from two mental health facilities in Province C, South Korea, which were cohort-isolated due to COVID-19. The inclusion criteria were voluntary informed consent and at least 1 year of employment as a mental health nurse prior to the cohort quarantine. Float nurses who had not worked before the cohort quarantine were excluded.

### 2.3. Data Collection

Data were collected through individual in-depth interviews from August 2021 to September 2021. This study collected data through purposive sampling until the results of the analysis of information-rich cases were saturated [21]. The in-depth interviews were conducted in a private space so that nurses could participate in the interviews without concerns for the leakage of their personal information. For participants who requested virtual interviews due to COVID-19, the interviews were conducted via Zoom. Each interview lasted from 1 h to 1 h and 20 min, and each participant underwent one to two interviews. Data were collected using a semi-structured interview questionnaire consisting of open-ended questions. The main question was “Tell me about your experience of providing patient care in a locked psychiatric unit that was cohort-isolated due to COVID-19”, and the interviewer asked the following follow-up questions to garner more in-depth data: “What challenges have you faced as you provided patient care in a locked psychiatric unit that was cohort-isolated due to COVID-19?” “What did you like about providing patient care in a locked psychiatric unit that was cohort-isolated due to COVID-19?” “What has changed in your patient care for patients in locked psychiatric units after the cohort quarantine experience?” The participants’ nonverbal cues, such as facial expressions, behaviors, and voice, were documented in the field notes for reference during data analysis.

### 2.4. Data Analysis

The collected data were analyzed as follows per the phenomenological method proposed by Colaizzi [22].

First, the researcher thoroughly read and analyzed the content of the interview transcripts to grasp the overall meaning of their experiences as well as their mood, such as tone of voice. Second, the transcripts were read once again to extract key statements that held significance or were repeated that were deemed to encapsulate the essence of the nursing experiences within cohort-isolated locked psychiatric units. Third, abstract meaning was derived from the identified key statements to identify the formulated meaning of participants’ experiences. Fourth, the meanings were combined and classified to produce 13 subthemes, and the subthemes were clustered into 5 themes. Fifth, the comprehensive findings were described using direct quotations from participants’ statements that aptly illustrated the identified themes. Sixth, the descriptions were thoroughly reviewed to identify the essence of participants’ experiences. Finally, to validate the results, the results were shown to a colleague with qualitative research experience and two research participants to confirm that the analysis results well reflect their experiences.

### 2.5. Rigor

To ensure the rigor of this study, efforts were made to enhance its credibility, transferability, dependability, and confirmability, as proposed by Lincoln and Guba [23]. First, to enhance credibility, the research assistant transcribed the participants’ language verbatim, which was then cross-checked by the researcher to minimize any omissions or distortions in the data. Second, for transferability, participants were purposefully selected using purposeful sampling [21]. The accuracy of the analysis results in reflecting participants’ experiences was verified by two research participants. Third, to establish dependability, data were analyzed in strict adherence to Colaizzi’s [22] method, and the overall study process and results were evaluated by one nursing professor with rich qualitative research experience. Fourth, to enhance confirmability, the researcher diligently recorded biases, assumptions, and reflections in a journal from the beginning to the end of the study so that researcher’s biases were eliminated and participants’ experiences were reflected in their entirety.

### 2.6. Ethical Considerations

This study was approved by the Institutional Review Board at C University (CBNU-202109-HR-0097) and conducted in accordance with the declaration of Helsinki. The principal researcher informed the participants that they had the freedom to withdraw from this study at any time and adequately explained the purpose and process of this study before obtaining a written consent form. The study data were coded to maintain anonymity and were stored in the principal researcher’s password-protected computer for use only for research purposes. All participants in this study were given a gift as a token of appreciation.

## 3. Results

Ten participants (eight female, two male) were enrolled. The mean age was 46.40 years. Eight were married and had children, and two were single. Their length of career in mental health nursing ranged from one to twenty-five years, with a mean length of career of 97.20 months. The participants provided care for patients with schizophrenia, chronic depression, dementia, alcohol use disorder, and neurodevelopmental disorder. Seven were staff nurses, and three were charge nurses. The mean duration of their care for patients in cohort-isolated locked psychiatric units was 51 days (Table 1).

The patient care experiences of nurses in locked psychiatric units that were cohort-isolated due to COVID-19 emerged as thirteen themes and five theme clusters (Table 2). 

### 3.1. Challenges Intensified by the Nature of Mental Disorders

Most of the participants stated that their challenges were intensified due to the nature of psychiatric units, in addition to the difficulties caused by the cohort quarantine. During the cohort quarantine, float nurses who had no rapport with the patients were only able to take charge of a limited number of nursing activities, so these nurses did not offer much practical help. Moreover, national guidelines absolutely did not reflect the reality in psychiatric units and thus were inapplicable in most cases. The nurses stated that the frequently updated guidelines escalated confusion. The participants also mentioned that their difficulties were further aggravated by repeated refusals to accept transfer of psychiatric patients with COVID-19 to medical-surgical units with a negative pressure room.

#### 3.1.1. Unprotected Exposure to Infection Risk

Most participants stated that they had difficulty educating psychiatric patients about cohort quarantine due to an infectious disease, use of protective suits, and need for PCR tests and asking for their cooperation due to their diminished cognitive function and communication skills. The participants described how the use of face masks, face shields, and protective suits hid their faces that were known to the patients, and the need for patients to take PCR tests every three days and wear protective suits when being transferred to another hospital due to COVID-19 infection further fueled their anxiety, often manifesting as physical resistance. The participants recounted instances where patients who vehemently resisted caused their masks and face shields to be removed and tore their protective suits, exposing the nurses to the risk of infection without protection.

“80% of the patients didn’t have the ability to engage in basic communications. Even when I explain, they don’t understand why they have to take the PCR test. So, every time they do the test, they just feel that they are in danger and start resisting full throttle. It makes me sweat, my protective suit is torn, it was just a nightmare.”(Participant 4)

“I explained the situation to the patient that was confirmed with COVID and was trying to put on the protective suit on him. The patient couldn’t understand and fought tooth and nail, and during this process, my face shield came off, my face mask came off, and my protective suit was all torn apart.”(Participant 7)

#### 3.1.2. Indivisible Workload

Most participants stated that having a rapport with the patients is critical to providing care for psychiatric patients. The additional float staff sent to make up for the short staffing did not have a rapport with these patients, so the patients did not cooperate with them. As a result, these float nurses also did not want to see the patients face-to-face, and they did not provide any practical help.

“Our patients are really high acuity, and if nurses don’t know each patient inside out like we do, then it’s gonna be hard on both the nurses and patients.”(Participant 2)

“When the float nurses go into patient rooms, they’re just like strangers to the patients, which causes them to act out or not cooperate. So eventually, we have to go in and take care of the patients.”(Participant 7)

#### 3.1.3. Inapplicable Guidelines

Most participants mentioned that personnel in charge of the management of mass outbreaks were dispatched from the Central Accident Response Headquarters and Central Disaster and Safety Headquarters when the cohort quarantine was begun. However, because they were not health care professionals and were not aware the situation of mental health hospitals, they were the ones asking the hospital staff about the current situation or they provided inapplicable guidelines. As a result, the guidelines changed several times a day, which caused confusion and frustration.

“The guidelines from the morning would change at night. They weren’t even feasible in mental health hospitals. It was so frustrating.”(Participant 4)

“The Central Disaster and Safety Headquarters sent personnel, and the only thing he did was to ask us what we’re doing, why we’re doing it, and just to criticize us. He did not give us any sort of practical guidelines. At first, we just dived in headfirst.”(Participant 7)

#### 3.1.4. Difficulty Transferring Patients

The participants mentioned that they had challenges because regular hospitals that had negative pressure rooms would not accept psychiatric patients confirmed with COVID-19 due to prejudice and fear of psychiatric patients.

“Patients confirmed with COVID had to be transferred to a different hospital, but none of the hospitals wanted to take them because they’re psychiatric patients. That was really hard.”(Participant 2)

“We have to send them to a hospital with a negative pressure room, but it’s really difficult to transfer psychiatric patients to other hospitals even if they have COVID.”(Participant 5)

### 3.2. Concerns Regarding Limited Mental Health Care

Most participants were nervous and frustrated about the narrower scope of mental health care provided to the patients during the cohort quarantine, where COVID-19 infection control precedes all activities, and the possible adverse consequences on patients’ psychiatric problems.

#### 3.2.1. Nervous about Not Being Able to Approach Patients Closely

Most participants were notified by their hospitals to minimize contact with patients during the cohort quarantine to reduce their risk of infection. They mentioned that they were not able to perform their normal frequent rounds or patient counseling as before the quarantine. As a result, they were concerned that patients might be at risk of adverse outcomes during a psychiatric episode because they could not address the issue immediately.

“We were told to provide the optimal care while minimizing contact but we were so confused because we don’t know how to do that.”(Participant 8)

“Because the focus was on COVID during the cohort quarantine, I felt that all the mental health nursing activities are put on the backseat.”(Participant 5)

“When patients are really stressed, they sometimes hurt themselves, so we need to round frequently. I was nervous because we couldn’t do that.”(Participant 2)

#### 3.2.2. Feeling Sorry for Patients in Double Confinement

Most participants felt sorry for patients in the locked psychiatric units because all they could do for them was to just witness their frustration during the COVID-19 quarantine. The normal activities that were allowed in locked units, such as staff-accompanied walks, visits from family, temporary passes to go out, and phone privileges, were all prohibited during the quarantine. Participants were concerned that the patients’ psychiatric symptoms would be exacerbated.

“Patients couldn’t leave their rooms at all or call their family or have visits from family. They were completely severed from the outside world, so the patients that had frequently interacted with their caregivers particularly had a hard time.”(Participant 6)

“Even before the cohort quarantine, the unit was a locked unit, so the patient couldn’t leave the unit by themselves. But they were still allowed to take a walk outside with a staff. But during the cohort quarantine, they could not leave the room at all, so they were very frustrated. I felt like if I kept them locked up, their psychiatric problems would get worse, not the coronavirus.”(Participant 7)

### 3.3. Escalating Stress

Most participants could not return home even after their shift due the cohort quarantine. They had to stay in the hospital for 24 h, and having no personal life like this was difficult. Moreover, their pay was markedly lower than that of nurses dispatched by the government, so they felt relatively deprived and unrecognized for their dedication.

#### 3.3.1. Having No Personal Life

Most participants mentioned that their inability to leave the hospital due to the cohort quarantine made them feel like their work shifts never ended. The loss of their personal life after work, such as spending time with family and engaging in recreational activities, was difficult.

“You know, you get some rest only if your shift ends and you get to go home. But this going home part was gone. Because I’m not going home, it felt like I’m working 24-7. This alone was hard.”(Participant 8)

“I couldn’t even go out to get things I need or go out for a walk and most of all, it was really difficult that I couldn’t be with my family.”(Participant 6)

#### 3.3.2. Relative Deprivation

Most participants stayed in the cohort quarantine spaces for their patients and colleagues, but their wages were so low that they felt their dedication was not recognized. Nurses dispatched by the government were only able to perform a limited scope of activities and thus did not work as much as they did, but these nurses were paid three to four times more than they were. This spurred a sense of relative deprivation.

“The float nurses were paid 3–4 times more than we were. If you think about it, we were paid 3–4 times less and did more work.”(Participant 6)

“Of course, we did what we did to protect our patients and we didn’t do that for the money. But the pay was so low to the point that we felt our dedication is not appreciated at all.”(Participant 7)

### 3.4. Bolstered Identity as a Mental Health Nurse

Most participants mentioned that being quarantined with the patients as a cohort allowed them to truly empathize with and understand the challenges faced by patients in a locked unit. The participants solidified their identity as mental health nurses by volunteering to handle difficult tasks, being considerate of their colleagues, and encouraging one another while undergoing similar challenges and being there for their patients.

#### 3.4.1. Stronger Rapport

Most participants were able to truly empathize with the challenges faced by patients in the locked units during the cohort quarantine, and they felt that patients were sincerely grateful to them for being with them during difficult times. Through this, participants felt that their therapeutic relationship with their patients was strengthened.

“Now we really understand why the patients are so irritated, so our attitude toward the patients really changed. When they get frustrated, we would try to persuade them to endure a little more and try to comfort them. We were able to care for them wholeheartedly.”(Participant 3)

“Before, the patients used to be mad at us, saying that what have you done for us, but they witnessed how we stayed with their 24-7 and run around in protective suits. They were really grateful and they tried to follow our directions as much as possible.”(Participant 8)

“I felt like we conquered this together with the patients. We thought about each other’s difficulties.”(Participant 5)

#### 3.4.2. Leadership That Leads by Example

Most participants shared that they and their colleagues served as one another’s support systems, volunteering to deal with difficult tasks and being considerate of one another. They described how they would step up and support one another in facing difficult tasks. They mentioned that they comforted hired caregivers who were nervous about the risk of infection by volunteering to handle difficult tasks themselves. As health care providers responsible for the well-being of patients in the units, these nurses showed remarkable dedication, caring for one another and steadfastly undertaking their roles in patient care.

“When some things had to be done at night or some difficult things had to be done, I volunteered to do them. And my colleagues started doing that also.”(Participant 3)

“When the cohort quarantine was first initiated, the hired caregivers were nervous and the entire caregiving staff said they’ll quit. We had to stand more firm. We persuaded the caregivers to overcome this challenge together, and fortunately, they were persuaded, and the situation was resolved.”(Participant 6)

#### 3.4.3. United by a Sense of Mission

Most of the participants mentioned that even in physically and mentally draining situations during the indefinite cohort quarantine, they found a sense of fulfillment as they witnessed patients becoming more emotionally stable through the challenges. They expressed that worrying for both the patients and colleagues who would struggle if they gave up kept them steadfast in difficult situations, motivating them to remain in their positions and do their best with the given situation.

“There’s this special thing with psychiatric patients. When I show empathy, I can feel their closed hearts opening up and can see that they loosen up. When I see the patients getting better, that makes me feel so good. I think I was able to endure because of that.”(Participant 6)

“I thought that my kids would be fine because my husband’s there. But I thought that the kids in the ward have no one else. And if I leave, my colleagues would have a hard time too. So even during these difficult times, I never thought about quitting and just leaving. I just did what I could do in that situation.”(Participant 7)

### 3.5. Witnessing Changes That Have Begun

After the cohort quarantine, participants recognized that their hospitals implemented several measures to improve systems and facilities, such as expanding waiting rooms that could house inpatients before the confirmation of COVID-19 and installing more sinks and hand sanitizers in every patient room. Moreover, they observed a positive change in the approach to infection prevention activities among patients and hired caregivers, who used to have little awareness of infection prevention activities and demonstrated poor adherence even at the beginning of the cohort quarantine.

#### 3.5.1. Beginning of Environmental Change

The participants noted improvements in the hospital environment immediately after the cohort quarantine, with an aim to reduce the risk of mass outbreaks. They highlighted that, because psychiatric patients are often hospitalized involuntarily and thus may not have COVID-19 results prior to admission, the hospital expanded waiting spaces for patients to stay in until the confirmation of negative COVID-19 test results. They also mentioned positive changes such as the installation of sinks and hand sanitizers in each room.

“The new admits stay in the waiting ward until negative COVID-19 test results are confirmed, and when the results come back negative, they are taken to their rooms.”(Participant 2)

“Now there’s a sink in every room, and so many hand sanitizers are available around the unit, so the patients washed their hands frequently too.”(Participant 3)

#### 3.5.2. Beginning of Behavioral Change

The participants stated that patients who had not been cooperative with infection prevention activities and hired caregivers who were not so eager for hygiene care for infection prevention now recognized the importance of infection prevention and were more cooperative with infection prevention activities after the end of the quarantine.

“After the cohort quarantine was lifted, the patients are so much more cooperative than before with things like hand washing, wearing a face mask, and ventilating the room more frequently.”(Participant 8)

“The hired caregivers now think disinfection and hygiene care as essential activities, so they do them even without we request them.”(Participant 7)

## 4. Discussion

This study aimed to explore the experiences of patient care among nurses in locked psychiatric units that were cohort-quarantined due to COVID-19 in depth and comprehensively.

In the first theme, “Challenges intensified by the nature of mental disorders”, divided into the subthemes “Unprotected exposure to infection risk”, “Indivisible workload”, “Inapplicable guidelines”, and “Difficulty transferring patients”, the participants mentioned that the difficulties stemming from the cohort quarantine were compounded by the distinctive challenges inherent to psychiatric hospital settings and the characteristics of individuals with serious mental illnesses. Patients with severe mental illnesses face difficulties in comprehending infection-related information due to cognitive impairments, making it challenging to implement infection-preventive actions [4]. Consequently, adhering to hygiene guidelines is challenging for these patients. As a result, nurses caring for patients with mental illnesses have an increased workload and risk of infection, which in turn results in an elevated infection risk among patients with mental illness, in a vicious cycle. It is crucial to reduce nurses’ workload and ensure their safety, even for patient safety purposes. However, amid the COVID-19 pandemic, when patients with mental health disorders may have experienced an exacerbation of their existing psychiatric symptoms, such as anxiety and depression [24], float nurses who had no rapport with the patients actually caused patients to act out and thus did not offer any practical help in reducing the nurses’ workload. In addition, frequent changes to COVID-19 guidelines intensified nurses’ emotional fatigue [10]. According to the national guidelines, patients confirmed with COVID-19 had to be treated in negative pressure rooms equipped with a system to maintain negative pressure to prevent airborne infection [14], but most mental health hospitals lacked negative pressure rooms [1]. Moreover, patients confirmed with COVID-19 were rejected by other hospitals due to their psychiatric conditions, which increased the nurses’ workload amid inadequate preparation for dangerous circumstances. The absence of tangible support amid escalated workloads could precipitate burnout among nurses, so specific policies that define roles and alleviate workloads are imperative [12]. To avoid a repetition of the chaos wrought by a new infectious disease outbreak in the future, proactive collaboration between mental health professionals and infection control experts is vital to proactively establish practical infection prevention guidelines for psychiatric units. Additionally, to ensure the safety and well-being of patients with mental illnesses during infectious disease crises due to prejudice against these patients and lack of negative pressure rooms, it is important to establish specialized infection units for patients with mental disorders [25].

In the second theme, “Concerns regarding limited mental health care”, divided into the subthemes “Nervous about not being able to approach patients closely” and “Feeling sorry for patients in double confinement”, participants stated that due to the prioritization of infection prevention over psychiatric care, face-to-face counseling, staff-accompanied walks, and family visits were restricted, and they were worried that this would exacerbate patients’ psychiatric problems. Because people with mental illnesses are more sensitive to stress compared to the general population, they may develop more amplified negative emotional responses to COVID-19, and their existing mental health conditions might relapse or deteriorate [26]. Restrictive measures, such as social distancing and quarantine, for preventing in-hospital infections are often challenging to implement in mental health hospitals considering the nature of existing inpatient treatments and patients’ recovery processes [1]. The guidelines and added duties introduced due to the risk of infection in clinical settings created conflicts in mental health nurses in terms of performing their professional roles [12]. However, strictly adhering to infection prevention guidelines is crucial for the protection of health care workers [27]. Frequent contact with patients with an infectious disease elevates health care professionals’ risk of infection and thus may lead to physical and emotional problems [28]. Moreover, individuals with mental disorders are at a heightened risk of infection compared to the general population due to their cognitive impairment, little awareness of dangers, and lack of capability to protect themselves [26]. Therefore, diverse non-face-to-face measures utilizing advanced technology should be developed and provided so that nurses can provide mental health care while adhering to infection prevention and management guidelines [25]. Virtual mental health services would be a promising option for addressing the issues of social isolation and loneliness among patients caused by the absolute prohibition of family visits and outside trips [24] and caregivers’ distress due to not being able to visit the patients for even a short time [29].

In the third theme, “Escalating stress”, divided into the subthemes “Having no personal life” and “Relative deprivation”, the participants described their hardships of having no personal life as a result of the cohort quarantine and the feeling of relative deprivation due to low wages despite having a significantly higher workload than float nurses. The participants felt isolated [12] and felt sorry for their families as their own and their families’ daily lives were restricted for prolonged periods during the quarantine [1]. The persistent intrusion into their personal lives, such as restrictions on their private lives and severed interpersonal relationships due to the infectious disease, exacerbated nurses’ stress [30]. Nurses who are quarantined and have to stay away from their families until their COVID-19-related work is over experience psychological problems such as worry, anxiety, pain, and loneliness due to their inability to fulfill their roles as family members [18]. Moreover, the participants indicated that low wages despite being assigned more demanding tasks compared to float nurses and often not being able to take their breaks further increased their stress [12]. Mental health nurses require higher levels of expertise and commitment compared to nurses in other fields [19], making it challenging to replace them with nurses lacking clinical experience in mental health. The longstanding issue of the shortage of mental health professionals was further exacerbated by the explosive spread of COVID-19 [7], leading to an excessive workload and stress among mental health nurses. Health care professionals play a crucial role in preventing, controlling, and caring for quarantined individuals and patients confirmed with infections as well as in public health. Thus, a shortage of health care professionals involved in patient care could lead to a collapse of the health care system [31]. To foster a safe mental health environment in the face of threats from other emerging infectious diseases in the future, there is a pressing need to establish a system that addresses the fundamental problems in mental health care settings in addition to providing programs such as mindfulness programs to alleviate burnout and promote stress management for nurses [7]. It is imperative to address the excessive workload of nurses and resolve the chronic shortage of nursing staff to ensure a good work-life balance [32]. Additionally, accurately calculating wages and establishing a commensurate compensation system for nurses is essential, as nurses face more serious emotional and physical problems [28] compared to other health care professionals while providing direct patient care at the patients’ bedside [12].

In the fourth theme, “Bolstered identity as a mental health nurse”, divided into the subthemes “Stronger rapport”, “Leadership that leads by example”, and “United by a sense of mission”, the participants described how they solidified their identities as mental health nurses and came to understand their patients and colleagues as they continuously spent time with them amid the challenges of the cohort quarantine. Empathy is the ability to understand situations and emotions from the perspective of other individuals and convey that understanding back to them. When nurses genuinely empathize with patients, it fosters trust, building a positive relationship with the patients [33]. While undergoing the cohort quarantine with the patients, the nurses were able to empathize with the patients’ challenges and communicate with them, which helped build a stronger rapport and strengthened their therapeutic relationship. Amid unfamiliar and challenging situations, participants demonstrated responsibility as nurses and led their organizations through the crisis, enhancing their leadership and fostering unity [12]. Nurses quarantined due to COVID-19 voluntarily helped others and gave each other strength to overcome difficult situations [18]. While the media portrayed nurses as heroic figures who sacrificially led the way in crisis situations [34], relying solely on nurses’ sense of duty is not the solution to ensuring the quality of nursing care in the long term [12]. To continually improve the quality of nursing and sustain nurses’ sense of duty even after the eradication of COVID-19, it is important to identify nurses’ challenges and relentlessly strive to ameliorate the hospital environment based on these findings.

In the fifth theme, “Witnessing changes that have begun”, divided into the subthemes “Beginning of environmental change” and “Beginning of behavioral change”, participants stated that their hospitals implemented several measures to improve their systems and facilities for infection prevention and that patients and hired caregivers demonstrated increased adherence to infection prevention activities immediately following the end of the cohort quarantine. Hospitals took steps to mitigate the risk of mass outbreaks by creating waiting rooms for patients for whom COVID-19 testing was complicated by their psychiatric symptoms [35]. Following mass outbreaks in the facilities, hospitals made institutional changes such as adjusting spacing between beds, limiting group programs, and implementing the mandatory installation of handwashing facilities [25]. To reduce the risk of mass outbreaks, it is important to provide education programs tailored to the cognitive abilities of individuals with mental disorders, so that they can acquire accurate information about infection prevention and act accordingly [1]. Similarly, hired caregivers who assist patients with mental disorders at their bedside should also be given infection prevention education to foster an environment that protects patients, caregivers, and health care staff against infections. Therefore, nurses need to assess the knowledge and behavioral levels of patients and caregivers regarding infection prevention and strive to establish and maintain feasible and realistic regulations and facilities that can enhance infection prevention activities. Hospitals should actively consider the opinions of nurses who work closely with patients when implementing regulations and facility enhancements for infection prevention.

This study confirmed the experiences of nurses who worked in locked psychiatric units that were cohort-quarantined due to COVID-19, providing a direction for improving the environment and systems of mental health institutions that can stably secure the safety of patients and medical staff in the event of a new infectious disease.

Nevertheless, this study has several limitations. First, the study participants were limited to nurses working in closed wards at two mental health institutions in Province C, South Korea. Second, only two out of the ten participants were male nurses, and the gender ratio of participants was biased toward one side. Therefore, the generalizability of the results of this study is limited. Based on the results of this study, we would like to make the following suggestions regarding follow-up research. First, there is a need to expand and conduct research targeting participants from various regions. Second, there may be differences between the experiences of psychiatric nurses in closed units of psychiatric institutions who are cohort-isolated due to COVID-19 and those of float nurses, so it is necessary to conduct research to compare these. Lastly, we propose a study that recruits a similar gender ratio of participants and compares the differences in experiences according to the participants’ gender.

## 5. Conclusions

The experiences of nurses in locked psychiatric units that were cohort-quarantined due to COVID-19 were classified as “Challenges intensified by the nature of mental disorders”, “Concerns regarding limited mental health care”, “Escalating stress”, “Bolstered identity as a mental health nurse”, and “Witnessing changes that have begun”.

Based on the findings of this study, we recommend the following strategies to mitigate the risk of mass outbreaks of infectious diseases in locked units in mental health hospitals.

First, mental health professionals should collaborate with infection control experts to establish feasible and practical infection prevention guidelines for locked psychiatric units and specialized infection control units for patients with mental disorders. Second, various non-face-to-face approaches using advanced technology should be developed so that nurses can provide mental health care in a safe environment while adhering to infection prevention guidelines. Third, an accurate calculation of wages and compensation systems for specialized work activities should be developed to address emotional difficulties among mental health nurses in order to address the issue of mental health nursing in the short term. Fourth, hospitals should actively listen to and reflect the opinions of mental health nurses when altering existing systems and facilities to ensure the prevention of mass outbreaks in locked psychiatric units.

## Figures and Tables

**Table 1 healthcare-11-02650-t001:** General characteristics of the participants (*n* = 10).

Participant	Sex	Age	Marital Status	Children	Psychiatric Ward Career (Month)	Diseases of Patients Cared for during Cohort Quarantine	Position	Cohort Quarantine Period: from December 2020 to February 2021
1	Male	48	Married	(2 children)	36	Schizophrenia, depressive disorder, dementia, alcoholism, etc.	nurse	45 days
2	Female	49	Married	(2 children)	40	Schizophrenia, intellectual disability, etc.	nurse	45 days
3	Female	55	Married	(2 children)	12	Schizophrenia, depressive disorder, dementia, etc.	nurse	45 days
4	Female	42	Married	(2 children)	240	Schizophrenia, depressive disorder, etc.	charge nurse	45 days
5	Female	39	Married	(1 child)	96	Schizophrenia, depressive disorder, dementia, alcoholism, etc.	charge nurse	75 days
6	Female	55	Married	(2 children)	300	Schizophrenia, intellectual disability, etc.	nurse	45 days
7	Female	35	Married	(2 children)	70	Schizophrenia, intellectual disability, etc.	nurse	45 days
8	Female	30	Single	X	80	Schizophrenia, depressive disorder, dementia, etc.	charge nurse	45 days
9	Male	39	Single	X	24	Schizophrenia, depressive disorder, dementia, alcoholism, etc.	nurse	75 days
10	Female	72	Married	(1 child)	74	Schizophrenia, depressive disorder, dementia, etc.	nurse	45 days

**Table 2 healthcare-11-02650-t002:** Themes and subthemes from psychiatric nurse interview responses.

Themes	Subthemes
Challenges intensified by the nature of mental disorders	Unprotected exposure to infection risk
	Indivisible workload
	Inapplicable guidelines
	Difficulty transferring patients
Concerns regarding limited mental health care	Nervous about not being able to approach patients closely
	Feeling sorry for patients in double confinement
Escalating stress	Having no personal life
	Relative deprivation
Bolstered identity as a mental health nurse	Stronger rapport
	Leadership that leads by example
	United by a sense of mission
Witnessing changes that have begun	Beginning of environmental change
	Beginning of behavioral change

## Data Availability

The data presented in this study are available upon request from the corresponding author. The data is not publicly available due to privacy restrictions.

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
