# Peer review of "Experiences of Caring for Cohort-Isolated Patients among Nurses in Locked Psychiatric Units"

_healthcare, 2023, doi:10.3390/healthcare11192650_

Round 1

Reviewer 1 Report

An et al reported an interesting manuscript that investigates the patient care experiences of nurses of a locked psychiatric 64 unit that had been cohort isolated due to COVID-19 in order to present foundational data to identify potential phenomena that may occur in locked psychiatric units during an infectious disease outbreak and help nurses to protect themselves and their patients from the infection. A few concerns that may need to be further revised:

1. Data collection: Inclusion and exclusion need to provide

2. Sample size may need to expand

3. This study, PMC10349601, needs to include and discuss 

4. In their discussion, each result needs to be mentioned and discussed extensively

5. Limitations of the study need to provide as well

Author Response

Comment 1: Data collection: Inclusion and exclusion need to provide

Response 1: Thank you for your comments.Page 2, line 78~82; Added inclusion and exclusion criteria when recruiting participants for data collection, as suggested.

Comment 2: Sample size may need to expand

Response 2: Thank you for your comments. Page 2, line 85-86: It was additionally stated in the data collection section that the sample size for this study was determined according to qualitative research method procedures.

Comment 3: This study, PMC10349601, needs to include and discuss 

Response 3: Thank you for pointing this out. Page 2, 56-59, Page 10, line 428-430, Page 11, line 463-465: As you suggested, ‘PMC10349601’ was included in this study and discussed in depth.

Comment 4: In their discussion, each result needs to be mentioned and discussed extensively

Response 4: Thank you for your comments. Page 9, line 363-368, 395-400, Page10, line 420-423, 451-455, Page 11, line 471-476: As you suggested, The discussion of this study has been modified to allow extensive discussion of each result.

Comment 5: Limitations of the study need to provide as well

Response 5: Thank you for your comments. Page11-12, line 492-507: As you suggested, Additional limitations of this study were presented.

Reviewer 2 Report

In particular, this is a study to prevent the occurrence of infectious diseases that may occur in the ward under the environmental conditions of cohort isolation due to COVID-19 and to faithfully perform the role of a nurse while simultaneously managing the health of patients.

Overall, the paper is well structured.

As a qualitative study, the purpose of the study and statistical analysis method are judged to be appropriate.

Sufficient consideration is being given based on research results.

Based on the research results, it is judged that policy recommendations are well presented.

Author Response

Comment 1: In particular, this is a study to prevent the occurrence of infectious diseases that may occur in the ward under the environmental conditions of cohort isolation due to COVID-19 and to faithfully perform the role of a nurse while simultaneously managing the health of patients.

Response 1: Thank you so much for your kind words of encouragement. We sincerely hope that this study can be helpful to patients and those who care for them.

Reviewer 3 Report

This is a solid and insightful study that will open the eyes of many people to the experiences of mental health professionals in the pandemic. I thought the results and discussion were organized well around theme clusters, and each theme was thoroughly and clearly explained. I thought the survey quotations were appropriate and well-placed. I'm not familiar with the Colaizzi method, but the general methodology as a qualitative study made sense. 

The only unclear part of the article was in Table 2. What does "patient's disease" mean? Does this mean the diseases that the nurse treated during the pandemic? Was there only one patient per nurse? Is just a statement of the nurse's specialty? This could be made clearer. And what is a negative pressure room? 

Overall, this was a very interesting read! I hope it has an impact on mental health practice. 

Author Response

Comment 1: The only unclear part of the article was in Table 2. What does "patient's disease" mean? Does this mean the diseases that the nurse treated during the pandemic? Was there only one patient per nurse? Is just a statement of the nurse's specialty?

Response 1: Thank you for pointing this out. ‘Patient’s disease’ refers to the disease of patients cared for by nurses in mental health institutions who were quarantined in cohorts due to COVID-19. Page 4, line 151: ‘Patient’s disease’ refers to the disease of patients cared for by nurses in mental health institutions who were quarantined in cohorts due to COVID-19. As you suggested, Table 1 has been modified to communicate clearly.

Comment 2: And what is a negative pressure room? 

Response 2: Thank you for your comments. Page 9, line 380-383: As you suggested, to clearly convey the meaning of this, we have added information about negative pressure rooms to the discussion section.

Reviewer 4 Report

The manuscript was interesting and well-written. The following comments can help the authors to improve it:

1-      It is better to choose the keywords based on the MeSH term.

2-      The number of the interviewees is limited. The authors could regard this study as a “case study” to resolve this problem, if all nurses had worked in a specific healthcare facility. Please add the number of healthcare facilities that nurses worked there.

3-      Regarding the phenomenological study and phenomenological method for data analysis, the authors need to provide adequate information about the reasons for choosing these methods.

4-      In Table 1, is it possible to use the terms “theme” and “subtheme” instead of current titles?

5-      I think it is better to change the order of tables 1 and 2. The results section can be started with the participants’ characteristics followed by the themes and their descriptions.

6-      In the discussion section, the authors can initially present a summary of findings and then discuss them.

7-      Please add research limitations to the manuscript.

Author Response

Comment 1: It is better to choose the keywords based on the MeSH term.

Response 1: Thank you for your comments. Page 1, line 22: As you suggested,

Keywords were modified based on MeSH term. 

Comment 2: The number of the interviewees is limited. The authors could regard this study as a “case study” to resolve this problem, if all nurses had worked in a specific healthcare facility. Please add the number of healthcare facilities that nurses worked there.

Response 2: Thank you for pointing this out. Page 2, line 78, 85-86: As you suggested, The mental health facilities where the nurses worked were additionally described, and the sample size was additionally stated to have been determined in accordance with qualitative research method procedures. 

Comment 3: Regarding the phenomenological study and phenomenological method for data analysis, the authors need to provide adequate information about the reasons for choosing these methods.

Response 3: Thank you for your comments. Page 2, line 48-54; In the introduction, I explained the rationale for choosing a phenomenological method for this study.

Comment 4: Table 1, is it possible to use the terms “theme” and “subtheme” instead of current titles?

Response 4: We agree with this comment. Page 4, line 153: Changed to “theme” and “subtheme” as suggested.

Comment 5: I think it is better to change the order of tables 1 and 2. The results section can be started with the participants’ characteristics followed by the themes and their descriptions.

Response 5: Thank you for pointing this out. Page 4, line 151, 153: As you suggested, The order of Tables 1 and 2 has been changed.

Comment 6: In the discussion section, the authors can initially present a summary of findings and then discuss them.

Response 6: Thank you for your comments. Page 9, line 363-368, Page 10, line 395-400, 420-423, Page 11, line 451-455, 471-476: As you suggested, The discussion has been modified.

Comment 7: Please add research limitations to the manuscript.

Response 7: Thank you for your comments. Page 11-12, line 492-507: We have added the limitations of this study as you suggested.

Round 2

Reviewer 1 Report

Thanks the authors for their acceptance of my comments and well addition revised. I believe the re²vised manuscript is now fine to go further action.